# A Better Maintenance Strategy, a More Sustainable Hanok: Towards Korean Traditional Public Facilities

**Jeong-Hyuk Jeong, Deuk-Youm Cheon and Seung-Hoon Han \*** 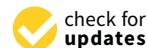

School of Architecture, Chonnam National University, 61186, Gwangju, Korea; rulldes@daum.net (J.-H.J.); dycheon@chonnam.ac.kr (D.-Y.C.)

**\*** Correspondence: hshoon@jnu.ac.kr; Tel.: +82-62-530-1646

**Abstract:** Currently, user demands for Hanok, the Korean traditional building type, are increasing in Korea, and their use as residences and accommodations are especially booming, while public facilities are rarely built in the style of Hanok these days. One of the most critical reasons for the issue is that Hanok lack usability and are difficult to maintain as a public facilities. Therefore, it is improvement of the usability of Hanok is needed for them to be accepted as public buildings and to set up the maintenance strategy for the style of the wooden structure. This research has defined public buildings in Hanok and classified them into three types according to structural standards. Then, this study analyzed the characteristics of each type, and suggested a checklist for Hanok usability, employing it to analyze the most recent exemplar Hanoks built as results in the governmental research project, monitored them in aspects of the facility management focused on the wooden structure and suggested strategies for sustainability by deriving eight factors in usability and another eight major types of defects to review the current status of the maintenance for the wooden structure. Finally, this study proposed the main direction for Hanok maintenance to establish its strategies for sustainability.

**Keywords:** Hanok; Korean traditional building; public facilities; Han style; usability; assessment system; maintenance strategy; sustainability

---

## 1. Introduction

Recently, the awareness of environmentally-friendly buildings, nature-friendly urban environment, and sustainable settlement for residents has increased when compared to the past. Hanok, Korean traditional buildings, are mainly used for residential facilities such as houses and accommodations, while inactively applied at public buildings in comparison [1]. Therefore, various attempts for Hanok to be a crucial type on non-residential architecture have been considered through empowering the usability of Hanok. In addition, maintenance and operation are quite important for the type of the wooden structure like Hanok to be more functional and sustainable as the multi-purposed facility for multiple users [2,3]. The purpose of this study is, therefore, to propose evaluation indexes for the usability of Hanok public buildings based on classifications of structural composition and to suggest the main strategies for maintenance operation with monitoring recently-built public facilities in the style of Hanok.

For this study, nine target buildings have been selected and analyzed with field surveys using professional devices and monitoring know-hows for a year from November 2017 to November 2018. All of them were built as prototyped testbeds of Korean traditional public facilities by the precedent R&D projects supported by Ministry of Land and Transport Affairs of Korean Government. The range of Hanok to be analyzed in this study is in the building style with Korean traditional characteristics and elements—especially merged with recent technologies, structures, and materials for contemporary

use. The scope of the public facility was also designated with a basis in precedent studies that defined it as the facility constructed by the budget of the government and open to the general public [4,5].

Therefore, this study has selected nine target buildings in accordance with the above two specifications to perform monitoring for their usability and maintenance status in order to reveal the main direction toward the Hanok sustainability. For this study, the target buildings were classified by the type of the structural formation with collecting data such as photographs, building references and architectural drawings as detailed summary of the cases. Then, the characteristics of the cases were analyzed in aspects of the spatial usability and the maintenance performance using the checklist and the onsite estimation. Finally, the concept of the maintenance strategy for Hanok has been suggested.

## 2. Characteristics of Korean Traditional Public Facilities

There are various classification methods to distinguish the types of public facilities in Hanok proposed by a few official institutions in Korea like Ministry of Land, Transport and Maritime Affairs; Ministry of Culture, Sports and Tourism; The Korean Housing Association, and so on. As a result of analyzing those criteria defining Hanok and new-Hanok or neo-Hanok, it was judged that the most appropriate classification standard is establishing three different types according to their structural formations and materials [6]. The public buildings in Hanok can be typed according to the above criteria as shown on Table 1.

**Table 1.** Types of the public facility in Hanok.

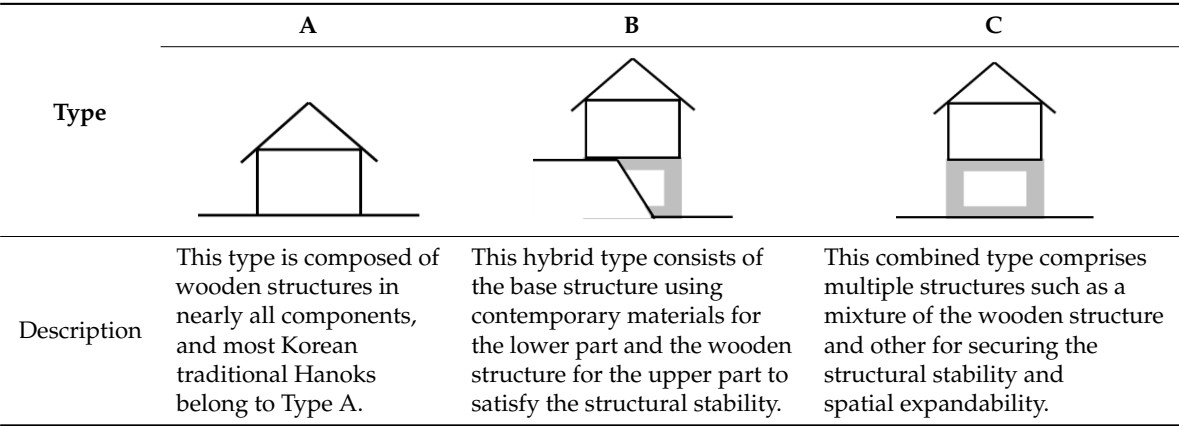

| Type | A | B | C |
|---|---|---|---|
| Description | This type is composed of wooden structures in nearly all components, and most Korean traditional Hanoks belong to Type A. | This hybrid type consists of the base structure using contemporary materials for the lower part and the wooden structure for the upper part to satisfy the structural stability. | This combined type comprises multiple structures such as a mixture of the wooden structure and other for securing the structural stability and spatial expandability. |

Type A normally utilizes the least amount of contemporary technologies and materials, and is the type most similar traditional Hanok. Type B consists of a base structure with current materials like reinforced concrete (RC) and a timber structure up above. Mostly, because of the level difference of the terrain, the underground is partly exposed and the lower structure is considered for structural stability and/or a supplementary space like storage, piloti, and so on [6,7]. Type C is composed of a variety of structures, materials, and relatively free formal formations, for example, RC as the lower structure, the ground layer with the wooden structure, and reformed plastic rooves in which combinations are widely accepted to support all creative Hanoks.

Among the three styles, Type A is closest to the most traditional form of Hanok. The surrounding environment affects the layout of its fundamental formation. Also, intermediate spaces connecting the rooms is one of its active characteristics taken from traditional Hanok. For example, as outlined on Table 2, the Literature Center at Eunpyeong Hanok Village located in Seoul, Korea uses many traditional elements such as Daechung as the main hallway, Maru as corridors, and yards defining the boundary between inside and outside fences. Another resultant prototype employing the mentioned R&D project is an energy-saving hotel at Gangneung Hanok Village in Gangneung. This facility was originally designed as a residential unit following the traditional rectangle plan, and it is now used as a hotel unit that is specialized to accept additional energy-saving components such as a photovoltaic roof while

keeping the traditional form [8,9]. Jishinjae Hanok in Youngin was the first prototype implemented for an experimental mock-up that functions as a laboratory to monitor building performance.

**Table 2.** Public Hanok facilities in Type A.

| Code | A-1 | A-2 | A-3 |
|------|-----|-----|-----|
| Building Name | Literature Center at Eunpyeong Hanok Village | Energy-saving unit at Gangneung Hanok Village | Jishinjae Hanok |
| Use | Exhibition, Office | Commercial, Hotel | Experimental mock-up |
| Location | Seoul | Gangneung, Gangwondo | Yongin, Gyeonggido |
| Year Built | 2013 | 2016 | 2013 |
| Picture | 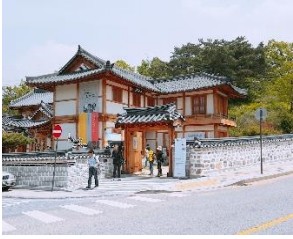 | 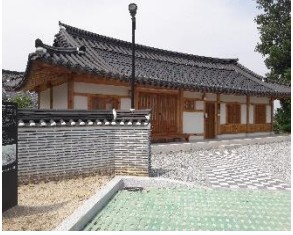 | 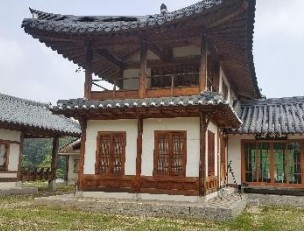 |

On the other hand, Type B is a public building that requires a wide space with no pillars due to its spatial characteristics, because a neck structure alone normally has a difficulty in securing a necessary space without pillars. For this reason, most of them use the contemporary construction materials like RC to form the base structure. They have often been used out of necessity and seem very passive in elevation to harmonize with the surrounding building components and adjacent environment [8–10]. All three cases in Type B summarized on Table 3 have a wooden structure for the upper part to demonstrate the traditional beauty of Hanok, and this combined structure is considered to satisfy the structural stability.

**Table 3.** Public Hanok facilities in Type B.

| Code | B-1 | B-2 | B-3 |
|------|-----|-----|-----|
| Building Name | Geumwa Kindergarten | Hanok Technology Center | Hotel units at Gangneung Hanok Village |
| Use | Education | Exhibition, Office | Commercial, Hotel |
| Location | Sunchang, Jeollanamdo | Suwon, Gyeonggido | Gangneung, Gangwondo |
| Year Built | 2016 | 2016 | 2016 |
| Picture | 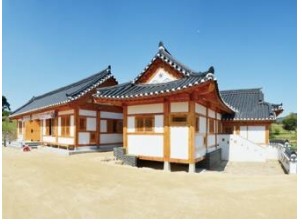 | 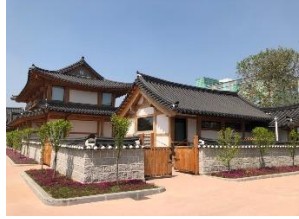 | 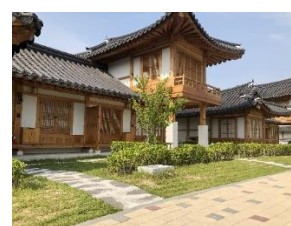 |

Type C accepts contemporary technologies and materials actively at most, and it is relatively important to harmonize the essential wood structure with other components as indicated on Table 4. In the case of the Community Center at Eunpyeong Hanok Village, the main space is composed of timber elements as the major structure and the base is a RC formation. Although two different building components may feel heterogeneous, the RC structure was finished with traditional materials, and this is why the two masses seem harmonized [10]. The Agricultural Experience Center in Naju expresses a

horizontally emphasized form with an intermediate space floor using glass doors that are a sort of contemporary materials, and the access floor of the Maintenance Center at Gangneung Hanok Village is also composed of modernized components that emphasize its functionality as a public facility.

**Table 4.** Public Hanok facilities in Type C.

| Code | C-1 | C-2 | C-3 |
|---|---|---|---|
| Building Name | Community Center at Eunpyeong Hanok Village | Agricultural Experience Center | Maintenance Center at Gangneung Hanok Village |
| Use | Community Service | Exhibition, Office | Office |
| Location | Seoul | Naju, Jeollanam-do | Gangneung, Gangwondo |
| Year Built | 2016 | 2016 | 2016 |
| Picture | 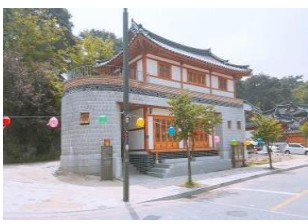 | 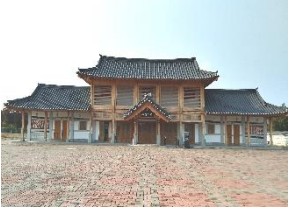 | 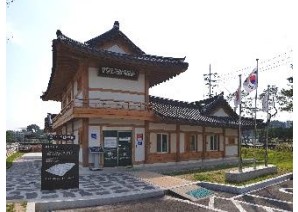 |

A field survey was conducted with 26 provincial Hanoks classified by nine target types introduced above. They comprise, in detail, 2 community buildings in Seoul, 2 public buildings in Gyeonggido (Suwon and Yongin); 1 administrative office and 19 hotel units in Gangwondo (Gangneung); and 2 public facilities in Jeollado (Naju and Sunchang) developed by the first and the second stages of Hanok Technology Development Research Project and supported by the Ministry of Land, Infrastructure and Transport Technology of Korean Government.

## 3. Framework of the Usability Evaluation and the Maintenance Monitoring

### 3.1. Research Methods for the Maintenance Monitoring toward Hanok

In societies, architecture has a duty to preserve the past and provide the possibility of building the present on the strength of culture and tradition [11]. For this study, monitoring methods for maintenance and management of Hanok have been utilized and they can be classified into two types: visual investigation and inspection using tools. Visual investigation is mainly performed through observations, but includes all five senses. This should be done only when the recognition of the phenomenon found on Hanok is very certain and the possibility of error for judgment is remarkably low. In some target cases for this research, therefore, measurement tools and supplementary devices have been used for reducing possible errors by visual inspections. Inspection using tools can be divided into hand tools such as a ruler, compass, divider, compass and plumb; and automated devices such as Schmidt hammer, crack detector, short-circuit tester, leveler, light detector, laser leveler and so on, depending on the simplicity and the precision of handling required for target Hanoks.

The inspection items are divided into two categories: spatial configuration to check the shape and position of each architectural element and performance to check whether Hanok maintains the function of its components. Firstly, the configuration status can be identified by visual inspections and/or mechanical measurements to detect problems such as deformation, movement, or departure among components. The performance check can be initially determined by observation or by the degree of beating in the case of intensity. However, inspections—such as insulation performance, humidity, and heat loss—that cannot be checked by the inspectors' sensory ranges, are carried out using automated devices.

### 3.1.1. Visual Inspection and Photography

Estimations by visual inspections should observe the phenomena such as deformation and movement compared with peripheral members to lower the possibility of error. Variations such as parallel twists, shifts, and departures are all to be known from the surrounding situation. In addition, defects of wooden elements such as deterioration may not be visually evident, so they should be checked by tapping or touching, because the state of the surface can only be observed visually. As shown on Figure 1, it will be an important method to check and clear the limitations visually even if other sensory organs are used, and to record the phenomenon through photography in order to confirm the third party of the experts for preparing alternatives.

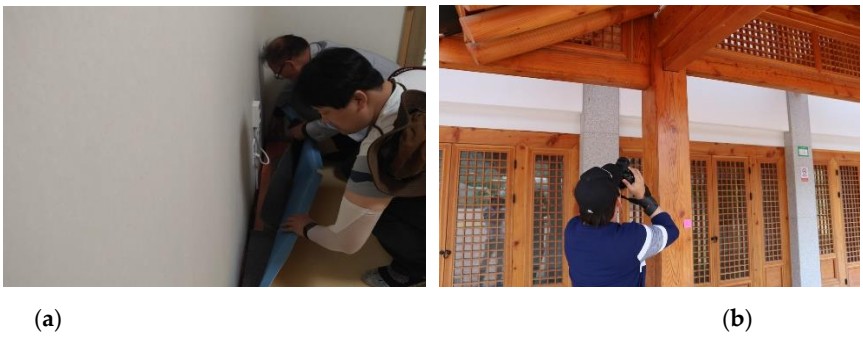

(**a**)　　　　　　　　　　　　　　　　　　　　　　　　　　　　　　(**b**)

**Figure 1.** Monitoring method for the Hanok maintenance: visual inspection and photography. (**a**) Tactile and visual inspection; (**b**) Recording through photography.

### 3.1.2. Visual Inspection by Hand Tools

Visual inspection can be difficult to distinguish properly if there is no subject to compare. Therefore, it is necessary to increase the reliability of inspection by using simple tools. The inspection by tools is basically an extension of the visual inspection, but the possibility of error is excluded as much as the tool is reliable.

For example, as shown on Figure 2, it is possible to check the inclination to the horizontal water level or to measure the inclined angle by putting a weight on the end of the column where the upper part of the Hanok column is tilted to one side. In the case of deformation of the wooden component, its value can be confirmed by using a taped ruler or Vernier calipers in comparison with a member such as a foundation stone that can hardly be deformed.

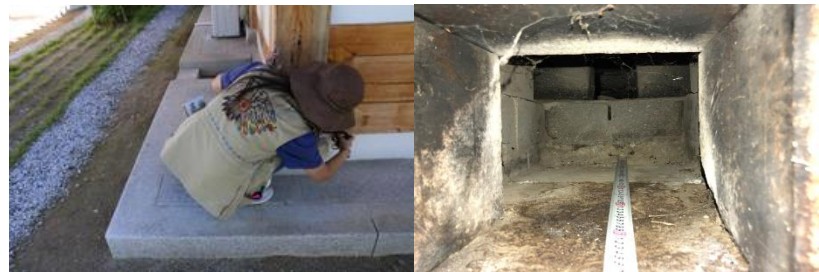

**Figure 2.** Monitoring method for the Hanok maintenance: visual inspection.

### 3.1.3. Tool Inspection by Automated Devices

Inspection by automated tools can be guaranteed to be highly reliable, but requires much more proficiency for using it. In order to obtain appropriate results for each circumstance and situation, the inspectors must understand how to use tools thoroughly and perform proficient operations to obtain reliable results as exemplified on Figure 3.

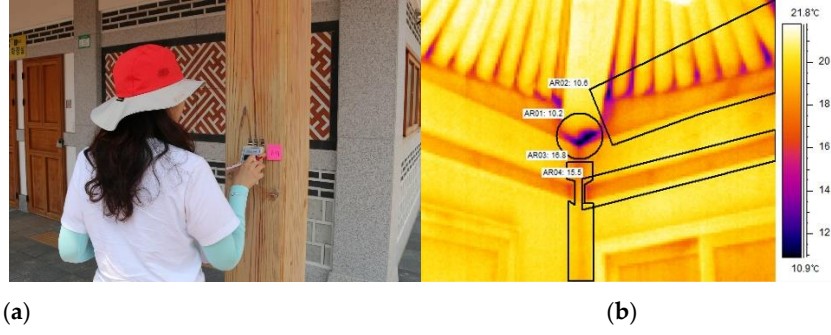

(**a**)　　　　　　　　　　　　　　　　　　　　　　　　　　(**b**)

**Figure 3.** Monitoring method for the Hanok maintenance: tool inspection by automated devices.
(**a**) Measurement using devices; (**b**) Analysis with infrared photography.

In case of the measurement for water content ratio, it is necessary to input information about wood species and set criteria for measurement under the same conditions. The point for the measurement is based on the center of the south at one meter above ground level for reference. In the case of the measurement for insulation performance of spatial members by infrared ray, the sample is to be collected based on the planner wall facing the outside air, and statistical processing should be performed by the ratio of the surface temperature difference (TDRi).

### 3.1.4. Maintenance Checklist for Field Investigations

This research has carried out the maintenance monitoring by using visual inspection, handy tools, automation devices, etc., checked whether there can be unusual matters through the checklist and attached photographs for each site as suggested on Figure 4. The maintenance checklist is an index of possible problems that can occur by site and spatial member or building component.

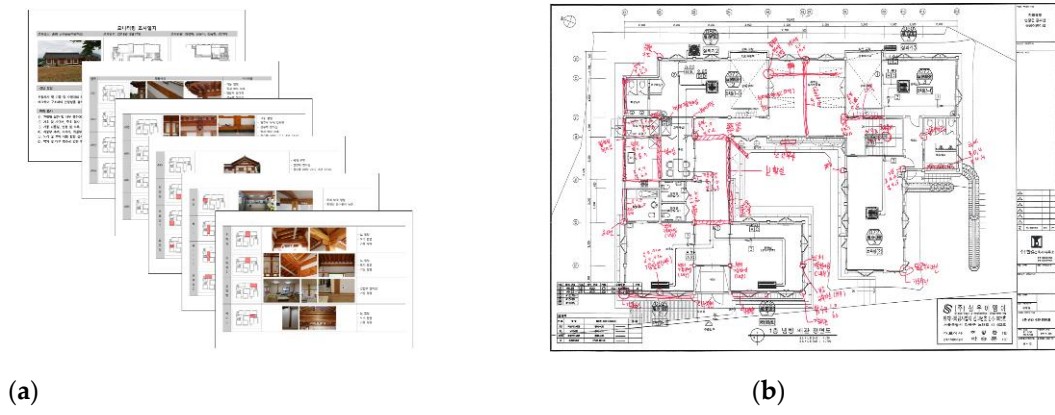

(**a**)　　　　　　　　　　　　　　　　　　　　　　　　　　(**b**)

**Figure 4.** Monitoring method for the Hanok maintenance: maintenance checklist. (**a**) Photographic record by section; (**b**) Damage checks through architectural drawings.

In the architectural features of each case, this study extracted the factors to be used for check of the usability. The contents are as shown on Table 5, and the assessment factors extracted from the architectural features can be classified into eight categories: elevation, human-oriented elements, territory, locality, circulation, barrier-free, maintenance, and communal space [8–10]. With these criteria, an onsite checklist questionnaire has been completed and prepared for checking of the Hanok usability. Enquiries may also be made and presented to the occupants, followed by data analysis and management as seen in most cases [12].

**Table 5.** Checklist for Hanok usability.

| Evaluation Items | Questions |
|---|---|
| Elevation | Is it possible to open the windows on the building and to ventilate naturally? |
| | Can you recognize original elements of Hanok on the building? |
| Human-oriented elements | Is there any stepped part in the room? |
| | Are the equipment installed considering the age of the user? |
| | Do furnishings have adequate heights for sitting or standing in the space? |
| Territory | Is there an intermediate space between the inner and the outer spaces? |
| | Is the height of the fence proper? |
| | Is the surrounding area separated by the intermediate space adjacent to the street? |
| Locality | Are the local environment and the building atmosphere harmonious? |
| | Have you used heterogeneous materials? |
| Circulation | Is there parking space in the suitable space? |
| | Is the building divided properly into separate buildings? |
| Barrier-free | Are there appropriate barrier-free elements? |
| | Are there any steps preventing the movement of handicapped people? |
| Maintenance | Is there any structural weakening or material changes from the original status? |
| | Does the maintenance manual or guidelines exist? |
| | Is it possible to contact the professional treatment in case severe defects are found? |
| Communal space | Are there public yards or plant gardens in the exterior space? |

As the project report of the precedent R&D project mentioned as the survey analysis, residents and users for Hanok pointed out that maintenance is the most important factor with 18% weights among the above usability items to guarantee the sustainability of the building [4,13]. They also insisted the needs of the maintenance manual and the systematical guidelines in order to determine the current status of the wooden structure and to find practical means of the treatment easily and promptly [14]. In this sense, this paper will serve as a fundamental guideline for maintaining Hanok that are built as a sensitive structure made of timbers and versatile materials. Survey results for target Hanoks have been recorded and summarized in Table 6, and the maintenance issues will be handled in detail with eight major defects that can be judged by normal users or residents.

**Table 6.** Sample survey results in the spatial usability for target Hanoks.

| Code | Feature |
|---|---|
| A-1 | - It is difficult to move wheelchairs because the space is narrow.<br>- Columns exist that prevent smooth movement within a unit space.<br>- It has a wide window and shows excellent performance in lighting and ventilation.<br>- Maintenance has been supported by the local government and kept well. |
| A-2 | - There is an intermediary space connecting the inner and the outer spaces such as the corridor (Maru) and the floor.<br>- The parking lot is located just in front of the building and often acts as an obstacle to the façade.<br>- The facility has been maintained by the local government and sustains good quality. |
| A-3 | - The conference room and the exhibition room in the basement require a large space.<br>- Underground RC is not exposed, so it is in harmony with the surrounding environment.<br>- Maintenance has not been performed successfully and many surficial defects can easily be found. |
| B-1 | - Kids' playroom is located in the basement and a larger space is needed.<br>- There are public gardens or vegetable fields in the exterior space.<br>- Maintenance work has regularly been executed and the building status is pretty fair. |

| Code | Feature |
|------|---------|
| B-2 | - By the outside space, Hanok elements such as stones and pillars are very visible.<br>- Due to the nature of exhibition space, residents need a large space without columns.<br>- RC structure on the lower part plays the role of foundation of Hanok.<br>- Maintenance has been supported by the local government and kept well. |
| B-3 | - In the facade, the underground space is exposed very passively and does not harm the tradition.<br>- Consideration should be given to a specific user because of steps in the space.<br>- The facility has been maintained by the local government and is still of good quality. |
| C-1 | - Due to the nature of the community facility, a large space for gathering is needed.<br>- The mezzanine space shown on the outside plays a role of the intermediate space.<br>- This facility used contemporary materials on its facade to express the intermediate floor.<br>- Maintenance has rarely been done and needs to be improved. |
| C-2 | - The building is divided into multiple levels of the floor and composed of contemporary spatial elements including a glass staircase, RC foundation, and so on.<br>- It has used heterogeneous materials, but there is harmony between two buildings.<br>- Maintenance should actively be performed on a regular basis. |
| C-3 | - It needs a large space in the upper floor not only for storing maintenance tools but for handling accommodation amenities.<br>- Barrier-free elements are not enough in spite of it being a public facility.<br>- Maintenance works have regularly been executed and the building status is pretty fair. |

*3.2. Evaluation for the Maintenance Monitoring of Hanok*

Some evaluation methods for the spatial usability devise a function analysis methodology that helps to identify 'performance of a user function' and refine the design procedure to 'fulfil a user requirement' by questioning what user needs are and how designers meet them [15,16]. The usability of the above buildings has been evaluated by onsite investigation using the checklist, and their maintenance status as one of the most important factors in the usability has been inspected by observing the defect types with visual observation, taking pictures, measuring the degrees of changes and distortions, enquiring how maintenance is done, and surveying the recognized defects from users and residents in target Hanoks [17]. The defects found on the buildings are divided into two categories from biological and abiotic causes, and after monitoring all 26 target Hanoks, a maintenance strategy could be suggested. Table 7 shows the types of defects by biological causes, and each item is summarized to share the onsite experiences.

**Table 7.** Defects due to biological causes.

| Defect Type | Frequency |
|-------------|-----------|
| Fungal generation and discoloration | 26 |
| Bluish changes | 26 |
| Putrefaction (Boohoo) | 0 |
| Termites | 0 |

3.2.1. Fungal Generation and Discoloration

Degradation due to biological causes is closely related to the weather condition. All of the target Hanoks were surveyed, and it is confirmed that degradation and discoloration mainly occur after the rainy season and happen not only on the wood structure but also on the outer wall. In addition, due to added moisture on the ground after rainfall, it could be observed intensively on the lower parts of the building components rather than on the upper parts. Distribution tendency of the discoloration shows that a greater passage of time results in a blacker surface as shown on Figure 5. Fungi belong to incomplete microorganisms and occur on surfaces of the building. They also tend to grow on the

surface without penetrating into the wood. However, there would be a heavy risk of discoloration if left unused for a long time.

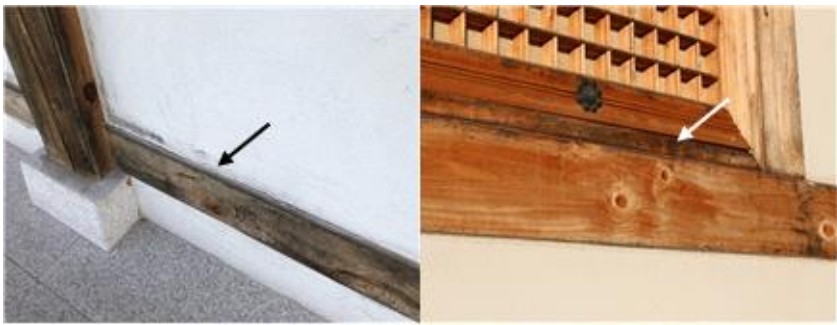

**Figure 5.** Hanok defects: fungal generation and discoloration.

### 3.2.2. Bluish Changes

Bluish changes are the most common defect among the degradation occasions caused by microorganisms. It is caused mainly by acanthaceous fungus among known microorganisms inhabiting the wood, normally, and by this situation, various discolorations occur depending on the species of timbers. Blue color does not deteriorate the physical properties, but could contaminate the appearance of the wood mostly causing a cosmetic problem as exemplified on Figure 6. This defect is one of the biological causes repeatedly observed from all target Hanoks during the field surveys. The investigation was conducted after the completion of demonstrative Hanoks, but there have been many cases in which the bluish change is generated when producing and processing timbers as well.

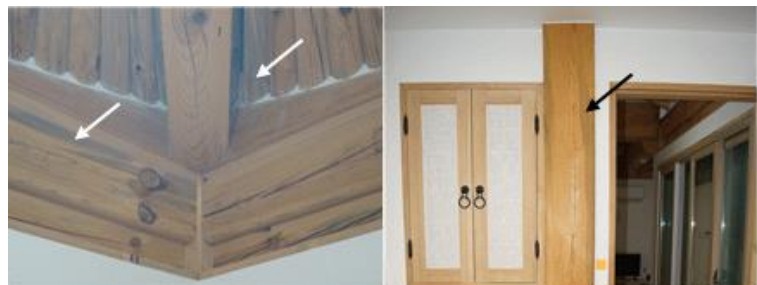

**Figure 6.** Hanok defects: bluish changes.

### 3.2.3. Putrefaction (Boohoo)

Boohoo is the phenomenon that the cell wall is broken down by the microorganism during the deterioration process of the wood, and can cause physical damage of the wood. This may weaken the structural strength of Hanok as a wooden structure and may cause the safety problems. No breakouts were observed from the target Hanoks during the surveyed period, and it seems to maintain its integrity in fair conditions, because they are just under five years old passed from its completion. However, a multi-storied traditional Hanok may encounter putrefaction issues in which the lower part of the column does not completely protect the rainwater from the eaves. This means that the bottom of the column has a greater chance of getting wet due to precipitation and has a significant effect on the occurrence of droughts. Observations should be continued and this issues will need focused attention in the future.

### 3.2.4. Termites

Termites are insects that ingest cellulose of plants and largely present in the form of wood, leaf piles, soil, and so on. These materials are originally important in maintaining the subtropical and tropical ecosystems. Korea is also experiencing global warming and the temperature is gradually

rising to become close to a subtropical climate, so there is a great concern about termites, especially for wooden building structures. Termite damage was not confirmed in all target Hanoks during the field study. This is because the target cases use concrete to form the building foundation mostly and lay cornerstones and a wooden structure is built on it. Primary physical control due to the concrete structure may be employed to prevent termites from entering the timber cells through the soil. Figure 7 shows sample maintenance sheets for monitoring Hanok defects mentioned previously through the field survey.

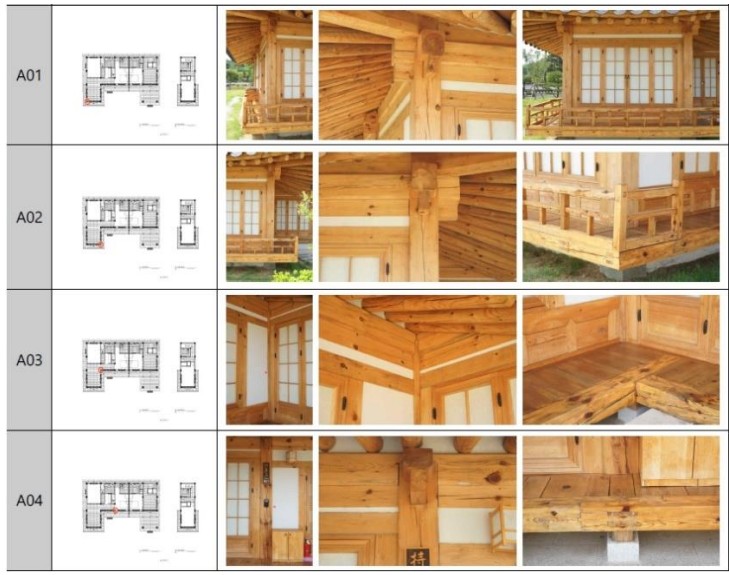

**Figure 7.** Sample maintenance sheet for the field survey for monitoring Hanok defects.

On the other hand, Table 8 indicates the status of them by abiotic factors, and each item has also been monitored and recorded to describe onsite conditions.

**Table 8.** Defects due to abiotic causes.

| Defect Type | Frequency |
|---|---|
| Parallel twists | 26 |
| Joint fissures | 26 |
| Wall cracks | 2 |
| Adhesive elution | 4 |

### 3.2.5. Parallel Twists

A typical wood defects observed commonly among all studied Hanoks is parallel twisting. A Hanok is a furniture-like building that supports the roof with a wooden structure, and the role of wood is very important. However, parallel twists could easily be observed on most building members using logs such as columns, girders, and beams. As described by the chapter of the timber structure in standard specifications for the cultural property repairs, the parallel twist should be controlled to be under 10% of the cross section. However, despite these standards, this type of defect becomes an element that hurts the beauty of Hanok. Particularly, the parallel twist on the end of the column for the joint is a main factor to lower the structural safety and is definitely the target of repair. In the case of target Hanok constructed with glued laminated woods, no parallel twists or ripples are generated, since the materials are provided by matching the regulated water content ratio. However, this defect was observed on horizontal materials like girders and beams using pine trees. The expansion of the margins or gaps on wooden components after construction normally occurs due to the drying

shrinkage of the material; it is suggested to use completely-dried timbers to prevent parallel twists. Table 9 indicates sample results in the water inclusion ratio (WIR) for a target Hanok.

**Table 9.** Sample results in the water inclusion ratio (WIR) for the target Hanok (B-3) on 7 June 2018.

| Space | Interior | | | | | Exterior 1 | | | | | |
|---|---|---|---|---|---|---|---|---|---|---|---|
| Column Number | S-1 | S-2 | S-3 | S-4 | S-5 | S-6 | S-7 | S-8 | S-9 | S-10 | S-11 |
| WIR (%) | 10.6 | 8.3 | 8.9 | 10.2 | 9.1 | 10.1 | 10.1 | 9.5 | 8.7 | 12.1 | 11.4 |
| Space | Exterior 2 | | | | | | | | | | |
| Column Number | S-12 | S-13 | S-14 | S-15 | S-16 | S-17 | S-18 | S-19 | S-20 | S-20 | - |
| WIR (%) | 11 | 9.9 | 10.4 | 14.3 | 14.2 | 11.3 | 15.6 | 15.3 | 10.6 | 10.5 | - |

### 3.2.6. Joint Fissures

In most target Hanoks, joint fissures from drying shrinkage of the timber could be found, and there is a widening in the area where the building components made of timbers like dry walls, columns, and lining materials are connected and fit each other as shown on Figure 8. As new-Hanok or shin-Hanok are being adapted to the lifestyle of contemporary people, the air tightness is getting more important in aspects of the energy performance, while traditional Hanok emphasized spatial breathability to assist natural air flow. Therefore, this defect not only lowers the airtightness for the contemporary Hanok, but also becomes an important cause of deterioration in performance.

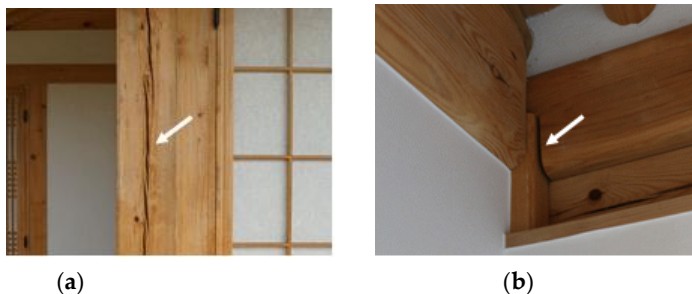

(**a**)                 (**b**)

**Figure 8.** Hanok defects: parallel twists and joint fissures. (**a**) Parallel twists; (**b**) Joint fissures

### 3.2.7. Wall Cracks

There is a big difference in the rate of deterioration of Hanok by resident existence in the space. In the case of a mock-up Hanok (A-3) that has not been occupied by residents for five years after construction, most of its drywalls have been cracked. Those of the testbed house (A-1) which is constantly used as a literature museum and an office, have not been cracked at all in the five years since it was built. On the other hand, in the case of the Maintenance Center at Gangneung Hanok Village (C-3), which has been under operation for two years after the completion of construction in 2016, as a special case, visible cracks have occurred on the drywall due to impacts and vibrations caused by the progress of the structural change for the inner wall. There is no significant effect on the performance of the drywall, but it can be improved through finishing repairs and may become better in aspects of the building aesthetics.

### 3.2.8. Adhesive Elution

It is advantageous for securing rigidity and preventing parallel twists to use the nature of the glued laminated wood with attaching and cumulating cut woods. For this reason, several target Hanoks utilize that kind of wooden component unlike most traditional ones, and therefore adhesive elution could found only from these structures as exemplified on Figure 9. The amount of glue or adhesives used for combining module timber elements was assumed to be small, but this defect

has polluted the surface of the building component with black discoloration and may be a negative aesthetic factor.

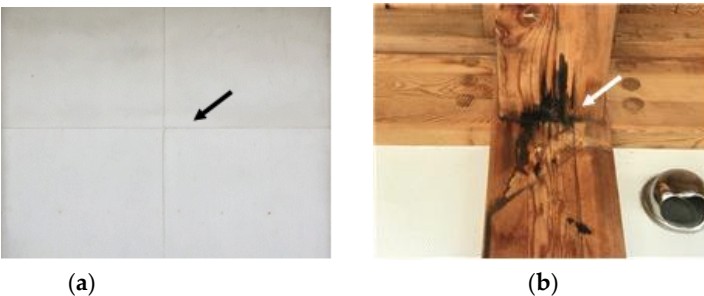

(**a**)　　　　　　　　　　　　　　　　　　　(**b**)

**Figure 9.** Hanok defects: parallel twists and joint fissures. (**a**) Wall cracks; (**b**) Adhesive elution

All 26 target Hanoks categorized in three different types are built by efforts with the capital of the host institution and supports from the subsidy of the government, and the ownership and management authority belong to the host institution. Figure 10 shows the current condition for Hanok maintenance performed on the site. As Hanok are based on the material of timbers, they must pay close attention to maintenance for better usability and sustainability in comparison to contemporary buildings. Interviews with the officers in host organizations and management experts also confirmed the need for a better maintenance system. Fundamental management is in progress to remove and repair defects occurred on target Hanoks, and it is based on daily management that removes surface contamination and proceeds with oil stain treatment to prevent pollution such as blue discoloration and fungi. In addition, irregular maintenance for the joint fissures or widening between the columns, the wall and/or joints have been treated with silicon. However, there was no separate official budget for regular, systematic maintenance and repairs.

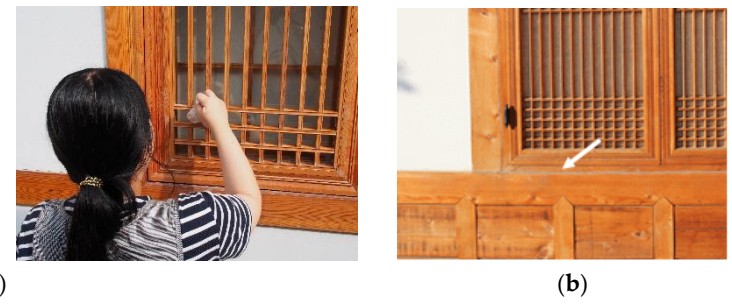

(**a**)　　　　　　　　　　　　　　　　　　　(**b**)

**Figure 10.** Current situation for Hanok maintenance performed on the site. (**a**) Ordinary manual maintenance; (**b**) Fixing fissures using silicon.

## 4. Conclusions

This study proposes a usability evaluation index and the maintenance guideline as its topmost factor for public facilities in Hanok. Based on the investigation, it was possible to identify characteristics of each type by categorizing public facilities from a structural point of view, and a Hanok maintenance framework could be suggested by analyzing those properties.

Furthermore, this study investigated the types of defects identified as the most important for Hanok by conducting field surveys of the maintenance status of target Hanoks; 9 representative cases and 26 target facilities in three categories have been studied. The most common defects are caused by fungi and bluish discoloration that are not related to the physical performance, but harmful to the aesthetics of Hanok. Especially, those pollutions are the most prominent defects found due to characteristics of the weather in Korea where rainy seasons occur every year.

In addition, parallel twists and the widening of pillars, walls, and trusses due to the drying shrinkage of the wooden components, not only damage the design of Hanok but also affect the building

performance in matters of structural safety, energy, and so on. On the other hand, joint fissures cause a decrease in the air tightness and deteriorate the insulation effect. Particularly, the segregation occurring in the defective parts may cause severe problems in the structural stability of Hanok. This is considered the most negative factor effecting sustainability.

Furthermore, a more restrictive standard for the water inclusion in timbers is required when considering the pattern of defects after completion, although the water content for the wooden structure is currently limited to 19% or less. Otherwise, it is impossible to secure the air tightness to improve the performance of Hanok or new-Hanok. The gap between the elements caused by drying shrinkage of the wood is difficult to repair or improve by technological achievements only due to the reality that the field conditions are always variable and optimized maintenance is needed for sustainability. Necessary regulations such as the water content ratio of the wood used for Hanok should also be implemented and applied to realize more sustainable Hanok.

**Author Contributions:** J.J., D.C. and S.H. conceived and designed the field surveys; J.J. and D.C. performed the evaluations; S.H. analyzed the data; J.J., D.C. and S.H. wrote the paper.

**Funding:** This research was funded by Ministry of Land and Transport Affairs of Korean Government. (Project No.: 18AUDP-B128652-02).

**Acknowledgments:** This research was supported by a grant from Urban Architecture Research Program (Technology Development of Design and Construction for Large-Space Hanok over 10 Meters, Development of Hanok Technology, Phase III).

**Conflicts of Interest:** The authors declare no conflict of interest.

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
