# Peer review of "A Better Maintenance Strategy, a More Sustainable Hanok: Towards Korean Traditional Public Facilities"

_buildings, doi:10.3390/buildings9010011_

Round 1
Reviewer 1 Report
the manuscript develops an approach of interest for the investigation of wood construction. Discussion of maintenance needs and procedures is important but the text does not ensure the minimum depth required by:
1- The state of knowledge on the subject is very small and more adequate to a technical framework for case study;
2 - The proposed guidelines have according to the authors support in previous research that carried out inquerito but that no evidence is presented that supports the reason of the checklist of table 5;
3 - The same happens with Table 6 that does not show any data that supports the presented maintenance issues;
4 - Regarding the references they appear in very small numbers even for the case study situation and should therefore be increased in particular from the theoretical theoretical introduction section of the research framework;
5 - It is necessary that the methodology of applied research and the discussion of the same face be communicated when reaching the findings.
The manuscript does not meet the conditions to be published in the present form.
Author Response
I really appreciate your comments and totally agree with your points of view, and I’ve revised my paper per your opinion; 1) more recent literature reviews have been added to provide sufficient background for the introduction, 2) research methodologies including the maintenance checklist have been added (pp.4-6), 3) supplementary materials have been adjusted to support the process of monitoring Hanok maintenance using the checklist on Table 5 and 6 accordingly, and 4) I would definitely try to investigate more about what should be done for the next paper due to the essential differences among target Hanoks that is expected to be performed till 2020. I would appreciate your warm consideration for this introductory article to be published in series of the researches for towards the heritage maintenance in Korea. Thank you very much once again.
Reviewer 2 Report
The paper is good and in my opinion it can be published. The only tip I would suggest is to add something like researching about what has been done in similar situations in the past. I'd say Trulli in Italy for example or in other countries where it is considered the value of the tradition linked to a building. It is for sure for next time paper. I do appreciate the 10 pages format, also that it is a synthetic standard for expressing ideas with a good scientific soundness without being prolix.
Author Response
I really appreciate your comments and totally agree with your points of view. I’ve added recent relevant references to provide sufficient background for the introduction. In addition, I would definitely try to investigate about what has been done in similar situations in the past for the next paper due to the essential differences among different principles of the management in the traditional building. Thank you very much once again.
Reviewer 3 Report
The relevant subject corresponds to the contents of the journal. The article is interesting and has a practical value. The information provided is useful.
However, the article has a major disadvantages.
It does not have an on-going piece of scientific articles:
1. There is no research methodology;
2. Not showed novelty;
3. No conclusions (only discussion);
4. A very poor literature review. The analyed articles should be referenced in international databases and dominate the latest publications.
At this time, it does not meet the requirement for a scientific article status. This article is an expert assessment or a technical description of the project.
I'm sorry, that I can not agree with this paper publiching. However, I urge the author to work seriously, to improve the article and to submit it again.
It's not useful to slip off the high level reached journal “Buildings” by publishing such articles, that have not yet been completed.
Author Response
I really appreciate your comments and totally agree with your points of view, and I’ve revised my paper per your opinion; 1) research methodologies have been added (pp.4-6) and supported the process of monitoring Hanok maintenance accordingly, 2) the novelty of this paper can, in my opinion, be found from many parts in the research, because precedent studies have not dealt with Hanok, Korean traditional architecture, in the subject of the sustainability and the maintenance yet. Please provide more detailed clues for this issue, 3) Conclusion has been included and revised, and 4) more recent literature reviews have been added to provide sufficient background for the introduction. In addition, I would definitely try to investigate more about what should be done for the next paper due to the essential differences among target Hanoks; it is expected to be performed till 2020. I would appreciate your warm consideration for this introductory article to be published in series of the researches for towards the heritage maintenance in Korea. Thank you very much once again.
Round 2
Reviewer 1 Report
The revised manuscript addresses the issues raised in the initial review.
Being an exploratory article and object of future development of the research, it meets the conditions to be published, but should be the object of English verification.
Reviewer 3 Report
Accept in present form